# Non Vanishing Gradients for Arbitrarily Deep Neural Networks: a Hamiltonian System Approach

**Clara L. Galimberti**
Institute of Mechanical Engineering
EPFL, Lausanne, Switzerland
clara.galimberti@epfl.ch

**Luca Furieri**
Institute of Mechanical Engineering
EPFL, Lausanne, Switzerland
luca.furieri@epfl.ch

**Liang Xu**
Institute of Mechanical Engineering
EPFL, Lausanne, Switzerland
liang.xu@epfl.ch

**Giancarlo Ferrari-Trecate**
Institute of Mechanical Engineering
EPFL, Lausanne, Switzerland
giancarlo.ferraritrecate@epfl.ch

## Abstract

Deep Neural Networks (DNNs) training can be difficult due to vanishing or exploding gradients during weight optimization through backpropagation. To address this problem, we propose a general class of Hamiltonian DNNs (H-DNNs) that stems from the discretization of continuous-time Hamiltonian systems. Our main result is that a broad set of H-DNNs ensures non-vanishing gradients by design for an arbitrary network depth. This is obtained by proving that, using a semi-implicit Euler discretization scheme, the backward sensitivity matrices involved in gradient computations are symplectic.

## 1 Introduction

Consider a $N$-layer deep neural network (DNN)

$$\mathbf{y}_{j+1} = \mathbf{f}(\mathbf{y}_j, \boldsymbol{\theta}_j), \quad j = 0, 1, \ldots, N-1, \tag{1}$$

where $\mathbf{y}_j \in \mathbb{R}^{n_j}$ and $\boldsymbol{\theta}_j \in \mathbb{R}^{n_{\theta_j}}$ denote the $j$-th layer feature vector and set of of parameters, respectively. Moreover, consider a training set for a classification task $\{(\mathbf{y}_0^k, c^k) \in \mathbb{R}^{n_0} \times \{1, \ldots, n_c\}\}_{k=1}^s$, where $\mathbf{y}_0^k$ are the input data and $c^k$ are the corresponding labels. In the framework of Deep Learning (DL), it is common practice to train the DNN (1) by minimizing the empirical risk, i.e.

$$\min_{\boldsymbol{\theta}_0, \ldots, \boldsymbol{\theta}_{N-1}} \quad \frac{1}{s} \sum_{k=1}^s \mathcal{L}(\mathbf{y}_N^k, c^k) \tag{2}$$

$$\text{s.t.} \quad \mathbf{y}_{j+1} = \mathbf{f}(\mathbf{y}_j, \boldsymbol{\theta}_j), \quad j = 0, 1, \ldots, N-1,$$

where $\mathcal{L}(\cdot, \cdot)$ is the loss function. When gradient descent methods are used, the update of the vectors $\boldsymbol{\theta}_j$ is performed using

$$\boldsymbol{\theta}_j^{(k+1)} = \boldsymbol{\theta}_j^{(k)} - \gamma^{(k)} \cdot \nabla_{\boldsymbol{\theta}_j^{(k)}} \mathcal{L}, \tag{3}$$

where $k$ is the iteration number and $\gamma^{(k)} > 0$ is the optimization step size at each iteration.

Even if DL has achieved remarkable success in various fields like computer vision, speech recognition and natural language processing [16, 25], the training of DNNs still presents several challenges such as the occurrence of vanishing or exploding gradients. Both situations are very critical as they imply that the learning process either stops prematurely or becomes unstable.

Workshop Paper at The Symbiosis of Deep Learning and Differential Equations Workshop at NeurIPS 2021.

At each iteration $k$ of (3), the gradient of the loss function with respect to the parameters needs to be calculated. By using the chain rule, the gradient for parameter $i$ of layer $j$ is computed through backpropagation [12] as:

$$\frac{\partial \mathcal{L}}{\partial \theta_{i,j}} = \frac{\partial \mathbf{y}_{j+1}}{\partial \theta_{i,j}} \frac{\partial \mathcal{L}}{\partial \mathbf{y}_{j+1}} = \frac{\partial \mathbf{y}_{j+1}}{\partial \theta_{i,j}} \left( \prod_{l=j+1}^{N-1} \frac{\partial \mathbf{y}_{l+1}}{\partial \mathbf{y}_l} \right) \frac{\partial \mathcal{L}}{\partial \mathbf{y}_N}. \tag{4}$$

The problem of vanishing/exploding gradients is related to the backward sensitivity matrices (BSM), i.e. the terms $\prod_{l=j+1}^{N-1} \frac{\partial \mathbf{y}_{l+1}}{\partial \mathbf{y}_l}$. Indeed, (4) implies that $\frac{\partial \mathcal{L}}{\partial \theta_{i,N-j}}$ vanishes if the term $\|\frac{\partial \mathbf{y}_N}{\partial \mathbf{y}_{N-j}}\|$ is very small. In other words, the training of a DNN might stop before achieving a good prediction performance. Vice-versa, if $\|\frac{\partial \mathbf{y}_N}{\partial \mathbf{y}_{N-j}}\|$ is very large, the derivative $\frac{\partial \mathcal{L}}{\partial \theta_{i,N-j}}$ becomes very sensitive to perturbations in the vectors $\frac{\partial \mathbf{y}_{j+1}}{\partial \theta_{i,N-j}}$ and $\frac{\partial \mathcal{L}}{\partial \mathbf{y}_N}$, and this can make the learning process unstable or cause overflow issues. Since the BSMs are related to the network depth, both problems are generally exacerbated when the number of layers $N$ is large [12].

Heuristic methods for dealing with these problems leverage subtle weight initialization or gradient clipping [12]. More recent approaches, instead, focus on the study of DNN architectures and associated training algorithms for which exploding/vanishing gradients can be avoided or mitigated *by design* [1, 18, 24, 17, 6, 26, 13, 14, 4, 21, 9].

For instance, in [1, 24, 17, 6] unitary and orthogonal weight matrices are used to control the magnitude of BSMs during backpropagation. Moreover, in [26, 18], methods based on clipping singular values of weight matrices are utilized to constrain the magnitude of BSMs. These approaches, however, require expensive computations during training [26, 18, 6], introduce perturbations in gradient descent [26, 18] or use restricted classes of weight matrices [1, 24, 17].

Recently, it has been argued that specific classes of DNNs stemming from the time discretization of Ordinary Differential Equations (ODEs) [13, 14, 4, 21, 9] are less affected by vanishing and exploding gradients. The arguments provided in [13] rely on the stability properties of the underlying continuous-time nonlinear systems for characterizing relevant behaviors of the corresponding DNNs obtained after discretization, suggesting to use DNN architectures based on dynamical systems that are *marginally stable*, i.e. that produce bounded and non-vanishing state trajectories. Examples are provided by first-order ODEs based on skew-symmetric maps, which have been used in [13, 4] for defining *anti-symmetric DNNs*, and by *Hamiltonian-inspired* DNNs in [13]. However, these approaches consider only restricted classes of skew-symmetric weight matrices or particular Hamiltonian functions, which limit the representation power of the resulting DNNs. Moreover, the behavior of BSMs arising in backpropagation has been analyzed only in [4], which however focuses on DNNs with identical weights in all layers and relies on hard-to-compute quantities such as kinematic eigenvalues [22, 2].

To address these open points, we propose a class of DNNs stemming from the symplectic discretization of Hamiltonian systems. We analyze the sensitivity dynamics in discrete-time and prove that the BSM is symplectic. As a result, the norm of the BSM cannot vanish, irrespective of the network depth and the choice of layer-varying weights.

## 2   H-DNN architecture

We propose H-DNNs as the time-discretization of continuous-time Hamiltonian dynamical systems, as per the standard definition given, for instance, in [23]:[1]

$$\dot{\mathbf{y}}(t) = \mathbf{J} \frac{\partial H(\mathbf{y}(t), t)}{\partial \mathbf{y}(t)}, \quad \mathbf{y}(0) = \mathbf{y}_0, \quad 0 \le t \le T, \tag{5}$$

where $\mathbf{y}(t) \in \mathbb{R}^n$ for all $t$, $\mathbf{J} \in \mathbb{R}^{n \times n}$ is skew-symmetric, i.e., $\mathbf{J} = -\mathbf{J}^\top$. The continuously differentiable function $H : \mathbb{R}^n \times \mathbb{R} \to \mathbb{R}$ is denoted as the *Hamiltonian function*. We consider,

$$H(\mathbf{y}(t), t) = [\tilde{\sigma}(\mathbf{K}(t)\mathbf{y}(t) + \mathbf{b}(t))]^\top 1_n, \tag{6}$$

---

[1]In a more general case, one can consider $\mathbf{J}$ to be a function of $\mathbf{y}$ and $t$, i.e. $\mathbf{J}(\mathbf{y}(t), t)$.

where $1_n$ denotes the vector of all ones of length $n$, $\tilde{\sigma} : \mathbb{R} \to \mathbb{R}$ is differentiable and applied element-wise, and its derivative $\sigma(\cdot) = \tilde{\sigma}'(\cdot)$ acts as the DNN's *activation function*. The system (5) can be rewritten as

$$\dot{\mathbf{y}}(t) = \mathbf{J}\mathbf{K}^\top(t)\sigma(\mathbf{K}(t)\mathbf{y}(t) + \mathbf{b}(t)), \quad \mathbf{y}(0) = \mathbf{y}_0, \quad 0 \le t \le T. \tag{7}$$

The H-DNN layer equations are obtained after selecting a discretization method. In this work, we use semi-implicit Euler (S-IE) [3] as it preserves useful geometric properties of (5). In particular, we will be interested in the preservation of the symplectic property of its flow. Assuming that the number of features $n \in \mathbb{N}$ is even,[2] we split the feature vector at each layer $j = 0, \ldots, N$ as $\mathbf{y}_j = (\mathbf{p}_j, \mathbf{q}_j)$ where $\mathbf{p}_j, \mathbf{q}_j \in \mathbb{R}^{\frac{n}{2}}$. Then, S-IE discretization of (7) leads to the layer equation

$$\begin{bmatrix} \mathbf{p}_{j+1} \\ \mathbf{q}_{j+1} \end{bmatrix} = \begin{bmatrix} \mathbf{p}_j \\ \mathbf{q}_j \end{bmatrix} + h \, \mathbf{J}\mathbf{K}_j^\top \sigma\left(\mathbf{K}_j \begin{bmatrix} \mathbf{p}_{j+1} \\ \mathbf{q}_j \end{bmatrix} + \mathbf{b}_j\right), \tag{8}$$

where $j = 0, 1, \ldots, N-1$ and $h = T/N$. In general, computing the updates $(\mathbf{p}_{j+1}, \mathbf{q}_{j+1})$ as per (8) involves solving a system of nonlinear equations, that is, $(\mathbf{p}_{j+1}, \mathbf{q}_{j+1})$ are not explicit functions of $(\mathbf{p}_j, \mathbf{q}_j)$. We refer the interested reader to [10, 20] for a discussion of deep learning with implicit layers equations. To make the updates (8) easily computable, one can further assume that

$$\mathbf{J} = \begin{bmatrix} 0_{\frac{n}{2}} & -\mathbf{X}^\top \\ \mathbf{X} & 0_{\frac{n}{2}} \end{bmatrix}, \ \mathbf{K}_j = \begin{bmatrix} \mathbf{K}_{p,j} & 0_{\frac{n}{2}} \\ 0_{\frac{n}{2}} & \mathbf{K}_{q,j} \end{bmatrix}, \ \mathbf{b}_j = \begin{bmatrix} \mathbf{b}_{p,j} \\ \mathbf{b}_{q,j} \end{bmatrix}, \tag{9}$$

which yields the explicit layer equations

$$\mathbf{p}_{j+1} = \mathbf{p}_j - h\mathbf{X}^\top \mathbf{K}_{q,j}^\top \sigma(\mathbf{K}_{q,j}\mathbf{q}_j + \mathbf{b}_{q,j}), \tag{10}$$

$$\mathbf{q}_{j+1} = \mathbf{q}_j + h\mathbf{X}\mathbf{K}_{p,j}^\top \sigma(\mathbf{K}_{p,j}\mathbf{p}_{j+1} + \mathbf{b}_{p,j}), \tag{11}$$

where one can first compute $\mathbf{p}_{j+1}$ through (10), while $\mathbf{q}_{j+1}$ is obtained as a function of $\mathbf{p}_{j+1}$ through (11).[3] Appendix A also presents the layer equations obtained when using forward Euler (FE) and it illustrates how H-DNNs generalize several architectures recently appeared in the literature [13, 5].

## 3 Non-vanishing gradients of H-DNNs

Our main result states that the H-DNNs described by (10)-(11) have BSMs lower-bounded in norm by the value 1 *irrespectively of the number of layers*. For this purpose, we first introduce the notion of symplectic matrix. Then, we show that, for any choice of weights in the form (9), the BSM obtained from S-IE discretization belong to this class. Finally we use symplecticity to show our main result.

**Definition 1 (Symplectic matrix)** *Let $\mathbf{Q} \in \mathbb{R}^{n \times n}$ be a skew-symmetric matrix. A matrix $\mathbf{M}$ is symplectic with respect to $\mathbf{Q}$ if $\mathbf{M}^\top \mathbf{Q}\mathbf{M} = \mathbf{Q}$.*

Symplectic matrices are usually defined [15] by assuming $n \in \mathbb{N}$ to be an even integer and $\mathbf{Q} = \begin{bmatrix} 0_{\frac{n}{2}} & I_{\frac{n}{2}} \\ -I_{\frac{n}{2}} & 0_{\frac{n}{2}} \end{bmatrix}$. In this respect, Definition 1 provides a slightly generalized notion of symplecticity.

**Lemma 1** *Consider the system (8) and assume that $\mathbf{J}$ has the block structure in (9). Then, the matrices $\frac{\partial \mathbf{y}_{k+1}}{\partial \mathbf{y}_k}$ for $j = 1 \ldots, N-1$ are symplectic with respect to $\mathbf{J}$, i.e.*

$$\left[\frac{\partial \mathbf{y}_{j+1}}{\partial \mathbf{y}_j}\right]^\top \mathbf{J} \left[\frac{\partial \mathbf{y}_{j+1}}{\partial \mathbf{y}_j}\right] = \mathbf{J}. \tag{12}$$

The proof of Lemma 1 can be found in Appendix B and is built upon Poincaré Theorem (see, e.g. Theorem 3.3 of Section VI in [15]) and the definition of extended Hamiltonian systems [7].

This result allows us to prove that the BSMs of H-DNNs are always lower bounded in norm by 1.

---

[2]This condition can be always fulfilled by performing feature augmentation [8].

[3]The layer equations (10)-(11) are analogous to those obtained in [13] and [5] by using Verlet discretization.

**Theorem 1** *Consider the H-DNN in* (8) *and assume that* **J** *has the block structure in* (9). *Then,*

$$\left\| \frac{\partial \mathbf{y}_N}{\partial \mathbf{y}_{N-j}} \right\| \geq 1 \,, \tag{13}$$

*for all* $j = 0, \ldots, N-1$, *where* $\|\cdot\|$ *denotes any sub-multiplicative norm.*

*Proof:* One has $\frac{\partial \mathbf{y}_N}{\partial \mathbf{y}_{N-j}} = \frac{\partial \mathbf{y}_N}{\partial \mathbf{y}_{N-1}} \frac{\partial \mathbf{y}_{N-1}}{\partial \mathbf{y}_{N-2}} \cdots \frac{\partial \mathbf{y}_{N-j+1}}{\partial \mathbf{y}_{N-j}}$. Then, by applying iteratively (12)

$$\left[ \frac{\partial \mathbf{y}_N}{\partial \mathbf{y}_{N-j}} \right]^{\top} \mathbf{J} \left[ \frac{\partial \mathbf{y}_N}{\partial \mathbf{y}_{N-j}} \right] = \mathbf{J}.$$

Hence, we have

$$\|\mathbf{J}\| = \left\| \left( \frac{\partial \mathbf{y}_N}{\partial \mathbf{y}_{N-j}} \right)^{\top} \mathbf{J} \left( \frac{\partial \mathbf{y}_N}{\partial \mathbf{y}_{N-j}} \right) \right\| \leq \left\| \frac{\partial \mathbf{y}_N}{\partial \mathbf{y}_{N-j}} \right\|^2 \|\mathbf{J}\| \,,$$

for all $j = 0, \ldots, N-1$. This inequality implies (13). ∎

The inequality (13) shows that the H-DNN architecture (10)-(11) guarantees non-vanishing BSMs by construction, irrespective of the network depth.

For a heuristic method to control the growth of the BSMs, we refer the reader to our work [11].

## 4 Numerical experiments

We provide a numerical validation[4] of the result in Theorem 1. We consider a classification problem over the 2D "Double moons" dataset (shown in Appendix C) and analyze the norm of BSMs during the training of a H-DNN with 32 layers (Figure 1a) and fully connected multilayer perceptron networks (MLPs) with 8 and 32 layers (Figure 1b). Detailed information about the architectures and the training parameters can be found in Appendix C.

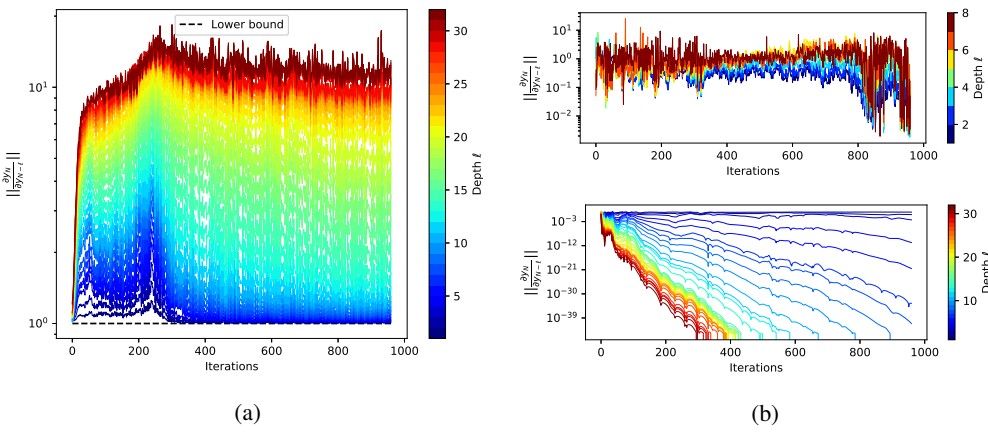

(a)                                        (b)

Figure 1: Evolution of the 2-norm of the BSM during the training of (a) a 32-layer H-DNN and (b) a multilayer perceptron network with 8 (top) and 32 (bottom) layers.

While the H-DNN and the 8-layer MLP achieve good performance at the end of the training (100% and 99.7% accuracy over the test set, respectively), the 32-layer MLP completely fails to classify the data (50% accuracy) due to rapidly vanishing gradients.

Figure 1a validates (13) since no BSM norm is smaller than 1 at any iteration.[5] However, this is not the case for the 32-layer MLP. Figure 1b (bottom) shows that, after 400 iterations, only a few gradient

---

[4]The open-source code is available at `https://github.com/DecodEPFL/HamiltonianNet`.

[5]Despite the BSM not vanishing, we verify that a stationary point $\boldsymbol{\theta}^{\star}$ such that $\nabla_{\boldsymbol{\theta}} \mathcal{L}(\boldsymbol{\theta}^{\star}) = 0$ is reached in approximately 350 iterations.

norms are different from zero, since $\left\| \frac{\partial \mathbf{y}_N}{\partial \mathbf{y}_{N-\ell}} \right\| \approx 0$ for $\ell = 7, \ldots, 32$. The main cause of such a premature stop of the learning is therefore directly linked to the phenomenon of vanishing gradients.

We refer the reader to Appendix D for further experiments demonstrating the potential of H-DNNs.

## 5  Conclusions

We introduced a class of H-DNNs obtained from the time-discretization of Hamiltonian dynamics and proved that H-DNNs stemming from S-IE discretization do not suffer from vanishing gradients. Although we limited our analysis to S-IE discretization, one can leverage the rich literature on symplectic integration [27] for defining even broader classes of H-DNNs with similar properties.

## Acknowledgments

Research supported by the Swiss National Science Foundation under the NCCR Automation (grant agreement 51NF40_180545).

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

## A  H-DNNs and existing architectures

We first introduce H-DNNs obtained using forward Euler (FE) discretization. Then, we show how existing architectures proposed in [13], [4] and [5] are included in our framework.

Consider the step size $h = T/N$. Then, the FE discretization of the ODE (7) leads to a $N$-layer H-DNN architecture defined as

$$\mathbf{y}_{j+1} = \mathbf{y}_j + h\,\mathbf{J}_j\,\mathbf{K}_j^\top \sigma(\mathbf{K}_j\mathbf{y}_j + \mathbf{b}_j)\,, \quad j = 0, 1, \ldots, N-1\,. \tag{14}$$

Having set two different architectures for H-DNNs, i.e. using S-IE (10)-(11) and FE (14), we list the existing architectures [13, 4, 5] and show that they are special cases of H-DNNs. These architectures stem from the discretization of Marginally Stable (MS) systems with constant weights across layers. Hence, we call them $\text{MS}_i$-DNN ($i = 1, 2, 3$).

**$\text{MS}_1$-DNN:** In [13], the authors propose to use Verlet integration method to discretize

$$\begin{bmatrix} \dot{\mathbf{y}}(t) \\ \dot{\mathbf{z}}(t) \end{bmatrix} = \sigma\left( \begin{bmatrix} 0_{\frac{n}{2}} & \mathbf{K}_0(t) \\ -\mathbf{K}_0^\top(t) & 0_{\frac{n}{2}} \end{bmatrix} \begin{bmatrix} \mathbf{y}(t) \\ \mathbf{z}(t) \end{bmatrix} + \begin{bmatrix} \mathbf{b}_1(t) \\ \mathbf{b}_2(t) \end{bmatrix} \right)\,,$$

where $\mathbf{y}, \mathbf{z} \in \mathbb{R}^{\frac{n}{2}}$. Then, the layer equations are given by

$$\begin{cases} \mathbf{z}_{j+1} = \mathbf{z}_j - h\sigma(\mathbf{K}_{0,j}^\top \mathbf{y}_j + \mathbf{b}_{j,1}) \\ \mathbf{y}_{j+1} = \mathbf{y}_j + h\sigma(\mathbf{K}_{0,j}\mathbf{z}_{j+1} + \mathbf{b}_{j,2})\,. \end{cases} \tag{15}$$

Note that (15) is an instance of H-DNN with S-IE discretization when $\mathbf{K}_j$ is assumed to be invertible and where we set $\mathbf{J}_j\mathbf{K}_j^\top = I_n$, and $\mathbf{K}_j = \begin{bmatrix} \mathbf{0} & \mathbf{K}_{0,j} \\ -\mathbf{K}_{0,j}^\top & \mathbf{0} \end{bmatrix}$ for all $j = 0, \dots, N-1$.

**MS$_2$-DNN:** In [13, 4], the authors propose to use FE to discretize

$$\dot{\mathbf{y}}(t) = \sigma(\mathbf{K}(t)\mathbf{y}(t) + \mathbf{b}(t)),$$

where $\mathbf{K}(t)$ is skew-symmetric $\forall t \in [0, T]$. Then, the layer equation is given by

$$\mathbf{y}_{j+1} = \mathbf{y}_j + h\sigma(\mathbf{K}_j\mathbf{y}_j + \mathbf{b}_j). \tag{16}$$

In this case, (16) is an instance of H-DNN with FE discretization when $\mathbf{K}_j$ is assumed to be invertible and where we set $\mathbf{J}_j\mathbf{K}_j^\top = I_n$ and $\mathbf{K}_j = -\mathbf{K}_j^\top$ for all $j = 0, \dots, N-1$.

**MS$_3$-DNN:** In [5], the authors propose to use Verlet integration method to discretize

$$\begin{bmatrix} \dot{\mathbf{y}}(t) \\ \dot{\mathbf{z}}(t) \end{bmatrix} = \begin{bmatrix} \mathbf{K}_1^\top(t) & 0_{\frac{n}{2}} \\ 0_{\frac{n}{2}} & -\mathbf{K}_2^\top(t) \end{bmatrix} \times \sigma\left( \begin{bmatrix} 0_{\frac{n}{2}} & \mathbf{K}_1(t) \\ \mathbf{K}_2(t) & 0_{\frac{n}{2}} \end{bmatrix} \begin{bmatrix} \mathbf{y}(t) \\ \mathbf{z}(t) \end{bmatrix} + \begin{bmatrix} \mathbf{b}_1(t) \\ \mathbf{b}_2(t) \end{bmatrix} \right),$$

where $\mathbf{y}, \mathbf{z} \in \mathbb{R}^{\frac{n}{2}}$. Then, the layer equations are given by

$$\begin{cases} \mathbf{y}_{j+1} = \mathbf{y}_j + h\mathbf{K}_{1,j}^\top\sigma(\mathbf{K}_{1,j}\mathbf{z}_j + \mathbf{b}_{j,1}), \\ \mathbf{z}_{j+1} = \mathbf{z}_j - h\mathbf{K}_{2,j}^\top\sigma(\mathbf{K}_{2,j}\mathbf{y}_{j+1} + \mathbf{b}_{j,2}). \end{cases} \tag{17}$$

Note that (17) is an instance of H-DNN with S-IE discretization where we set

$$\mathbf{K}_j = \begin{bmatrix} 0_{\frac{n}{2}} & \mathbf{K}_{1,j} \\ \mathbf{K}_{2,j} & 0_{\frac{n}{2}} \end{bmatrix} \quad \text{and} \quad \mathbf{J}_j = \begin{bmatrix} 0_{\frac{n}{2}} & I_{\frac{n}{2}} \\ -I_{\frac{n}{2}} & 0_{\frac{n}{2}} \end{bmatrix}.$$

In [13] and [5], the DNNs MS$_1$ and MS$_3$ are called *Hamiltonian-inspired* in view of their similarities with Hamiltonian models, although a precise Hamiltonian function for the corresponding ODE has not been provided. Moreover, note that the Verlet discretization used by the authors coincides with S-IE.

We highlight that a necessary condition for the skew-symmetric $n \times n$ matrix $\mathbf{K}_j$ to be invertible is that the size $n$ of input features is even.[6] If $n$ is odd, however, one can perform input-feature augmentation by adding an extra state initialized at zero to satisfy the previous condition [8].

## B  Proof of Lemma 1

The proof of Lemma 1 is built upon the result of Theorem 3.3 of Section VI in [15] and the definition of extended Hamiltonian systems [7]. The former proves that the numerical flow of a *time-invariant* Hamiltonian system with $\mathbf{J} = \begin{bmatrix} 0 & I \\ -I & 0 \end{bmatrix}$ is symplectic. The latter will allow us to embed the study of a time-dependent Hamiltonian function into the time-independent case by defining an extended phase space of dimension $n + 2$ instead of $n$.

We study the Hamiltonian system (5) in the *extended* phase space [7], i.e., we define an extended state vector $\tilde{\mathbf{y}} = (\mathbf{p}, \mathbf{q}, \varepsilon, t)$,[7] an extended interconnection matrix $\tilde{\mathbf{J}} = \begin{bmatrix} \mathbf{J} & 0_{n\times 2} \\ 0_{2\times n} & \mathbf{\Omega} \end{bmatrix}$, $\mathbf{\Omega} = \begin{bmatrix} 0 & -1 \\ 1 & 0 \end{bmatrix}$ and an extended Hamiltonian function

$$\tilde{H} = H(\mathbf{p}, \mathbf{q}, t) + \varepsilon, \quad \text{such that} \quad \frac{d\varepsilon}{dt} = -\frac{dH}{dt}. \tag{18}$$

Note that the extended Hamiltonian system defined by $\tilde{H}$ is time-invariant by construction, i.e. $\frac{d\tilde{H}}{dt} = 0$. Then, following Theorem 3.3 in Section VI of [15], it can be seen that $\frac{\partial\tilde{\mathbf{y}}_{j+1}}{\partial\tilde{\mathbf{y}}_j}$ is a symplectic matrix, i.e. it satisfies

$$\left[\frac{\partial\tilde{\mathbf{y}}_{j+1}}{\partial\tilde{\mathbf{y}}_j}\right]^\top \tilde{\mathbf{J}} \left[\frac{\partial\tilde{\mathbf{y}}_{j+1}}{\partial\tilde{\mathbf{y}}_j}\right] = \tilde{\mathbf{J}}. \tag{19}$$

---

[6]For a $n \times n$ skew-symmetric matrix $\mathbf{A}$ we have, $\det(\mathbf{A}) = \det(\mathbf{A}^\top) = \det(\mathbf{A}^{-1}) = (-1)^n\det(\mathbf{A})$. If $n$ is odd, then $\det(\mathbf{A}) = -\det(\mathbf{A}) = 0$. Thus, $\mathbf{A}$ is not invertible.

[7]Note that permuting the elements of $\tilde{\mathbf{y}}$, the state vector can be re-written as $(\tilde{\mathbf{p}}, \tilde{\mathbf{q}})$ where $\tilde{\mathbf{p}} = (\mathbf{p}, \varepsilon)$ and $\tilde{\mathbf{q}} = (\mathbf{q}, t)$.

Next, we show that (19) implies symplecticity for the BSM of the original time-varying system (8). Considering the Hamiltonian (18), we obtain the S-IE layer equations for the extended Hamiltonian dynamics as per:

$$
\begin{cases}
\mathbf{p}_{j+1} = \mathbf{p}_j - h\mathbf{X}^\top \frac{\partial H}{\partial \mathbf{q}}(\mathbf{p}_{j+1}, \mathbf{q}_j, t_j), \\
\mathbf{q}_{j+1} = \mathbf{q}_j + h\mathbf{X}\frac{\partial H}{\partial \mathbf{p}}(\mathbf{p}_{j+1}, \mathbf{q}_j, t_j), \\
e_{j+1} = e_j - h\frac{\partial H}{\partial t}(\mathbf{p}_{j+1}, \mathbf{q}_j, t_j), \\
t_{j+1} = t_j + h.
\end{cases}
\tag{20}
$$

Then, we differentiate each of the equations of the system (20) with respect to each subvector of $\tilde{\mathbf{y}}_j = (\mathbf{p}_j, \mathbf{q}_j, e_j, t_j)$ and we rearrange the terms. We obtain[8]

$$
\frac{\partial \tilde{\mathbf{y}}_{j+1}}{\partial \tilde{\mathbf{y}}_j}\left(I_n - h \begin{bmatrix} H_{pp} & H_{qp} & 0 & H_{tp} \\ 0 & 0 & 0 & 0 \\ 0 & 0 & 0 & 0 \\ 0 & 0 & 0 & 0 \end{bmatrix} \begin{bmatrix} \mathbf{J} & 0 \\ 0 & \mathbf{\Omega} \end{bmatrix}^\top\right) =
$$

$$
\left(I_n + h \begin{bmatrix} 0 & 0 & 0 & 0 \\ H_{pq} & H_{qq} & 0 & H_{tq} \\ 0 & 0 & 0 & 0 \\ H_{pt} & H_{qt} & 0 & H_{tt} \end{bmatrix} \begin{bmatrix} \mathbf{J} & 0 \\ 0 & \mathbf{\Omega} \end{bmatrix}^\top\right), \quad (21)
$$

where

$$
H_{xy} = \frac{\partial H(\mathbf{p}_{j+1}, \mathbf{q}_j, t_j)}{\partial x \partial y},
$$

and $x, y$ indicate any combination of two variables in the set $\{p, q, t\}$. It remains to verify that (21) implies symplecticity of $\frac{\partial \mathbf{y}_{j+1}}{\partial \mathbf{y}_j}$, i.e. it satisfies (12) for:

$$
\frac{\partial \mathbf{y}_{j+1}}{\partial \mathbf{y}_j} = \begin{bmatrix} \frac{\partial \mathbf{p}_{j+1}}{\partial \mathbf{p}_j} & \frac{\partial \mathbf{q}_{j+1}}{\partial \mathbf{p}_j} \\ \frac{\partial \mathbf{p}_{j+1}}{\partial \mathbf{q}_j} & \frac{\partial \mathbf{q}_{j+1}}{\partial \mathbf{q}_j} \end{bmatrix}.
$$

By denoting $\mathbf{\Gamma} = \left(I_n - h\begin{bmatrix} H_{pp} & H_{qp} \\ 0 & 0 \end{bmatrix}\mathbf{J}^\top\right)$ and $\mathbf{\Lambda} = \left(I_n + h\begin{bmatrix} 0 & 0 \\ H_{pq} & H_{qq} \end{bmatrix}\mathbf{J}^\top\right)$, where $\mathbf{\Gamma}$ and $\mathbf{\Lambda}$ are invertible for almost every choice of step size $h$, the part of (21) concerning $\frac{\partial \mathbf{y}_{j+1}}{\partial \mathbf{y}_j}$ reads as

$$
\frac{\partial \mathbf{y}_{j+1}}{\partial \mathbf{y}_j}\mathbf{\Gamma} = \mathbf{\Lambda}.
$$

The above implies

$$
\frac{\partial \mathbf{y}_{j+1}}{\partial \mathbf{y}_j}^\top \mathbf{J} \frac{\partial \mathbf{y}_{j+1}}{\partial \mathbf{y}_j} = \mathbf{J} \iff \mathbf{\Lambda}^\top \mathbf{J} \mathbf{\Lambda} = \mathbf{\Gamma}^\top \mathbf{J} \mathbf{\Gamma},
$$

where the equality $\mathbf{\Lambda J \Lambda} = \mathbf{\Gamma J \Gamma}$ can be verified by direct inspection. We conclude that $\frac{\partial \mathbf{y}_{j+1}}{\partial \mathbf{y}_j}$ is a symplectic matrix.

It is worth remarking that similar conclusions can be obtained by performing the analysis in continuous-time before selecting a discretization scheme. Hence, one can prove that the continuous-time counter-part of H-DNNs has non-vanishing BSMs.

## C   Implementation details

The DNN architectures and training algorithms are implemented using the PyTorch library.[9]

We use the "Double moons" dataset shown in Figure 2. It consists of 16,000 2-dimensional points, that are equally split to obtain the train and test datasets.

The layer equations of the H-DNNs are described in (10)-(11) where we set $\mathbf{X}$ to be the identity matrix. For the MLPs, their layer equation is given by:

$$
\mathbf{y}_{j+1} = \sigma(\mathbf{K}_j \mathbf{y}_j + \mathbf{b}_j),
$$

---

[8]To improve readability, the dimension of the zero matrices has been omitted.
[9]https://pytorch.org/

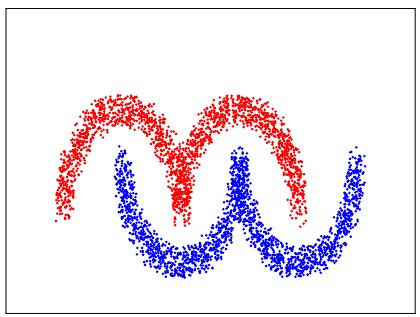

Figure 2: Double moons dataset with its corresponding labels (red and blue).

with trainable parameters $\mathbf{K}_j$ and $\mathbf{b}_j$. We use $\tanh(\cdot)$ as activation function for both architectures. Moreover, we complement each of them with a soft-max output layer.

Training is performed using coordinate gradient descent, i.e. a modified version of stochastic gradient descent (SGD) with Adam ($\beta_1 = 0.9$, $\beta_2 = 0.999$) [13] and minibatches of size 125. We use standard cross-entropy as the loss function $\mathcal{L}$ to minimize in (2), together with the regularization term

$$\alpha \, R_K(\mathbf{K}_{0,\dots,N-1}, \mathbf{b}_{0,\dots,N-1}) + \alpha_N R_N(\boldsymbol{\theta}_N). \tag{22}$$

The term $R_K$ is defined as $\frac{h}{2} \sum_{j=1}^{N-1} \left( \|\mathbf{K}_j - \mathbf{K}_{j-1}\|_F^2 + \|\mathbf{b}_j - \mathbf{b}_{j-1}\|^2 \right)$ which favours smooth weight variations across consecutive layers [13, 5]. The term $R_N(\cdot)$ refers to a standard $L_2$ regularization for the output layer. We set $\alpha_N = 1 \times 10^{-4}$ and $\alpha = 5 \times 10^{-4}$.

Following [13], in every iteration of the algorithm, the optimal weights of the output layer are computed given the last updated parameters of the hidden layers. Then, a step update of the hidden layers' parameters is performed by keeping fixed the output parameters. The training consists of 50 epochs and each of them has maximum 10 iterations to compute the output layer weights. The learning rate, or optimization step size as per $\gamma$ in (3), is set to $2.5 \times 10^{-2}$.

## D   Experiments with MNIST dataset

We evaluate our methods on a more complex example: the image classification benchmark MNIST.[10] The dataset consists of $28 \times 28$ digital images in gray scale of hand-written digits from 0 to 9 with their corresponding labels. It contains 60,000 training examples and 10,000 test examples.

Following [13], we use a network architecture consisting of a convolutional layer followed by a Hamiltonian DNN and an output classification layer. The convolutional layer is a linear transformation that expands the data from 1 to 8 channels, and the network output is a vector in $\mathbb{R}^{10}$ representing the probabilities of an image to belong to each of the 10 classes.

We compare the performance of MS$_1$-DNNs (see Appendix A) [13] and H-DNNs (using S-IE discretization) with 2, 4, 6 and 8 layers, and $\tanh(\cdot)$ as activation function. We set $h = 0.4$ for MS$_1$-DNNs and $h = 0.45$ for H-DNNs.

The training consists of 40 epochs with mini-batches of size 100. For the optimization algorithm we use SGD with Adam [19]. The learning rate, or optimization step size as per $\gamma$ in (3), is initialized to be $0.04$ with a decay rate of $0.8$ at each epoch.

We utilize cross-entropy loss function $\mathcal{L}$ and we add a regularization term given by

$$\alpha \, R_K(\mathbf{K}_{0,\dots,N-1}, \mathbf{b}_{0,\dots,N-1}) + \alpha_{L_2} R_{L_2}(\boldsymbol{\theta}).$$

The term $R_K$ is defined as $\frac{h}{2} \sum_{j=1}^{N-1} \left( \|\mathbf{K}_j - \mathbf{K}_{j-1}\|_F^2 + \|\mathbf{b}_j - \mathbf{b}_{j-1}\|^2 \right)$ which favours smooth weight variations across consecutive layers [13, 5]. The term $R_{L_2}(\cdot)$ refers to a standard $L_2$ reg-

---

[10]http://yann.lecun.com/exdb/mnist/

Table 1: Classification accuracies over training and test sets for the MNIST example when using $MS_1$-DNN (see Appendix A) [13] and H-DNN architectures. A convolutional layer and an output layer are added before and after each DNN. The first row, corresponding to 0 layers, refers to a network with a single convolutional layer followed by an output layer.

| Number of | $MS_1$-DNN | | H-DNN | |
|---|---|---|---|---|
| layers | Train | Test | Train | Test |
| 0 | 93.51% | 92.64% | - | - |
| 2 | 99.20% | 97.95% | 98.90% | 97.60% |
| 4 | 99.11% | 98.23% | 99.51% | 98.28% |
| 6 | 99.58% | 98.10% | 99.53% | 98.25% |
| 8 | 99.80% | 98.26% | 99.38% | 98.35% |

ularization over all the DNN trainable parameters. For $MS_1$-DNN, we set $\alpha = 1 \times 10^{-3}$ and $\alpha_{L_2} = 1 \times 10^{-3}$. For H-DNN, we set $\alpha = 4 \times 10^{-3}$ and $\alpha_{L_2} = 2 \times 10^{-3}$.

In Table 1, we summarize the accuracies of the considered networks on train and test data. The first row of the table provides, as a baseline, the results obtained when omitting the Hamiltonian DNN block, i.e., when using only a convolutional layer followed by the output layer. We observe that both $MS_1$-DNN and H-DNN achieve similar performance. Note that, while the training errors are almost zero, the test errors are reduced when increasing the number of layers, hence showing the benefit of using deeper networks. Moreover, these results are in line with test accuracies obtained when using standard convolutional layers instead of H-DNNs [5].

