# OpenReview forum: "Non Vanishing Gradients for Arbitrarily Deep Neural Networks: a Hamiltonian System Approach"
_NeurIPS.cc/2021/Workshop/DLDE — DLDE Workshop -- NeurIPS 2021 Poster_

### Official Review · Reviewer_sb2u · 2021-10-01
**Overall this is a manuscript that clearly describes a methodology that has the property of lower bounding the magnitude of gradients.**

**Confidence:** 3

**Review:**

I question the novelty of this methodology, however please consider that I am not familiar with the state-of-the-art of this topic.

The clear and short explanation of the H-DNN architecture is appreciated.

The methodology clips the hidden layers derivatives. Which in consequence enables the flow of error defined by the loss function. This is the major claim and impact that this manuscript has. I am interested in follow up works of this technique, to see how one can also upper bound the gradients in order to prevent exploding gradients using related techniques.

I think that this work presents a nice methodology to tackle the problem. However, the authors should in future work address the other techniques, that are also available in previous work, and compare them with this method. Nonetheless, the preliminary results confirm the hypothesis drawn at the beginning of the document.

I feel that the contributions of this work are original and clear, however the significance remains to be addressed by the authors with further experiments, comparison and analysis in order to publish.

Suggestions:

Consider changing the structure of the introduction section: you start by describing a problem, which is how gradients flow through a neural network. However the major topic is not addressed right at the beginning, only at the second page.

Since the problem addressed is specifically the impact your approach has on vanishing gradients, exploding gradients should not be referred in the manuscript

Related work that tackle the problem of vanishing and exploding gradients is somehow missing. It would be appreciated to reference techniques, such as batch normalization, regularization, skip connections, among others. All the previously, directly and/or indirectly, tackle this problem.

A reset of Table 1 in order to respect publication ready tables would be appreciated.

Corrections:

"In particular, we will be ..." -> "In particular, we are ..."

"$\sum_{k=1}^{s}$" -> "$\sum_{k=1}^S$" consider changing number of instances to caps $S$

"validationof" -> "validation of"

**Score:**

3: Good paper

---

### Official Review · Reviewer_yeh2 · 2021-10-10
**DNN as discretized Hamilton system**

**Confidence:** 3

**Review:**

In this work, the authors proposed a class of DNN that can be viewed as a numerical approximation of the Hamiltonian system, as opposed to previous work where the system is often only a numerical approximation of first order ODE.

A few comments

- The authors could include some analysis of computational costs of this methods, especially as opposed to the vanilla MLP.
- I think theorem 1 only guarantees that the gradient will not vanish, however, the authors should also comment on how to prevent the gradient explosion problem implied by theorem 1 (e.g. using some activation with decaying first order derivatives). I think discussion in this perspective is lacking in this work, but it could be some drawbacks as the DNN goes deeper.

Pros
- Firm theoretical presentations about the architecture.
- Fairly detailed description of previous works and motivations.

Cons
- The literature review in the introduction section is to long, compared to the fairly limited length of this extended abstract. The author could have been spent 1/4 of these paragraphs describing the numerical results.

**Score:**

3: Good paper

---

### Official Review · Reviewer_rCU4 · 2021-10-12
**Nice use of the analogy between discretized DEs and neural architectures; unclear implications for applications**

**Confidence:** 4

**Review:**

The authors present a class of neural network architectures that are guaranteed to avoid the vanishing gradient problem. This property is obtained (and proven) by deriving the architecture as the discretization of a Hamiltonian-governed dynamical systems using a sympletic Euler method. The proposed architectures are strict generalizations of several recently presented architectures, which are also based on discretizations of dynamical systems. This is a clever approach to solving a very fundamental problem in deep learning using insights from the analogy of deep learning architectures with differential equations. Thus, the paper is certainly very relevant to the DLDE workshop, and I recommend it be accepted.

As the other reviewers have commented, although the authors discuss the vanishing and exploding gradient problems in tandem, it is not clear that their architecture prevents exploding gradients from occurring. Myself and the other reviewers all appear to have been led to expect that the H-DNN architecture would do something to mitigate exploding gradients. Although the title of the submission clearly addresses only vanishing gradients, this aspect of the text may be worth clarifying more explicitly in the future.

My immediate reaction to the submission was to wonder whether the proposed H-DNN architecture is of practical utility. The authors provide some evidence that H-DNNs can still solve problems: they show that the architectures can solve the Double Moons problem as well as a simple MLP; they mention that gradient-based training for that problem did converge upon a fixed point in the loss landscape; and they show that H-DNN can solve MNIST as well as the related MS1-DNN architectures. Nonetheless, the impact of the work would be greatly improved in the authors could find an example of a non-trivial problem on which the H-DNN architecture exhibits some clear advantage over alternatives.

The authors have done a good job of motivating the importance of overcoming the vanishing gradient problem, and of the theoretical advantages of the H-DNN method to alternative methods of overcoming vanishing gradients. Is there perhaps a setting in which vanishing gradients are particularly problematic and on which H-DNNs can achieve better final performance than other methods? The natural setting for this would likely be where extremely deep architectures have substantial advantages; see the experiments in the original ResNet paper (https://arxiv.org/abs/1512.03385) or later works on extremely deep ResNets, perhaps. Alternatively, perhaps an analysis of computational cost can show a practical advantage to H-DNN?

The disappointing alternative hypothesis would be that, although H-DNNs are guaranteed to avoid vanishing gradients, for some reason they derive no applied benefit from this behaviour. This would seem remarkable unlucky to me, in which case I expect that the authors should be able to find a compelling example of impressive H-DNN performance in practice.


**Score:**

4: Very good paper

---

### Decision · Program_Chairs · 2021-10-16

**Decision:**

Accept (Poster)

**Comment:**

Reviewers have recommended acceptance. The approach to avoid vanishing gradients seems very promising.